# Genome Reorganization during Erythroid Differentiation

**DOI:** 10.3390/genes12071012

**Published:** 2021-06-30

**Authors:** Anastasia Ryzhkova, Nariman Battulin

**Affiliations:** 1Institute of Cytology and Genetics SB RAS, Laboratory of Developmental Genetics, 630090 Novosibirsk, Russia; ryzhkova@bionet.nsc.ru; 2Department of Natural Sciences, Novosibirsk State University, 630090 Novosibirsk, Russia

**Keywords:** erythroid differentiation, 3D genomics, TADs, chromatin organization

## Abstract

Hematopoiesis is a convenient model to study how chromatin dynamics plays a decisive role in regulation of cell fate. During erythropoiesis a population of stem and progenitor cells becomes increasingly lineage restricted, giving rise to terminally differentiated progeny. The concerted action of transcription factors and epigenetic modifiers leads to a silencing of the multipotent transcriptome and activation of the transcriptional program that controls terminal differentiation. This article reviews some aspects of the biology of red blood cells production with the focus on the extensive chromatin reorganization during differentiation.

## 1. Introduction

For a long time chicken erythrocyte nuclei have been a classic model object in numerous studies on the chromatin arrangement in interphase cells. Organization of a chromatin fibre and different levels of its compaction, from a 146 bp nucleosome-level units to large structurally distinct chromatin compartments, were studied using nucleated erythrocytes [1,2,3,4,5]. In particular, there has been a long debate regarding the existence of a 30 nm fibre as one of the basic levels of chromatin organization in living cells: this structure has been detected in chicken erythrocytes and starfish sperm chromatin, both on isolated chromatin fibers and in the native nucleus using Cryo- EM and synchrotron X-ray scattering methods [6,7,8,9,10]. It is now clear that chromatin organization of erythrocytes is an extremely rare case. Their nuclei become gradually more condensed during maturation, which results in extensive transcriptional silencing. Chromatin reorganization involves histone modifications, e.g. deacetylation, DNA demethylation, selective histone release and probably other yet unrevealed factors. At the moment, a lot of data has accumulated on the chromatin structure of late erythroid cells, which is almost completely transcriptionally inactive, indicating a fundamentally different organization of their nuclei, making them somewhat unique.

## 2. Erythroid Differentiation

During mammalian erythropoiesis, erythroid progenitors proliferate and undergo stepwise morphological changes, producing a pool of nuclear-free reticulocytes [11]. Erythropoiesis occurs in a series of steps, starting from a hematopoietic stem cell (HSC) giving rise to the sequential stages of erythroid precursors—a slowly proliferating burst forming unit-erythroid (BFU-E) and a rapidly dividing colony forming unit-erythroid (CFU-E). Over the next three days, each CFU-E produces 30–50 enucleated reticulocytes that are released into a bloodstream. Erythropoietin (Epo), a cytokine secreted by the kidney and hepatic perisinusoidal cells, is a major regulator of RBC production. Epo binds to cognate receptors on CFU-E cells and activates the JAK2 protein kinase with several downstream signaling pathways. These stimulate the proliferation and differentiation of CFU-E. They differentiate into large proerythroblasts with a nucleus occupying up to 80% of the cell volume and well-distinguished nucleoli. Further development from proerythroblast to reticulocyte, known as terminal erythroid differentiation, involves four to five rapid mitotic divisions, with the subsequent formation of basophilic, polychromatic and orthochromatic erythroblasts [12]. During these maturation stages, the cell size decreases from 25 µm to 8 µm in humans, and the size of the nucleus is reduced [13].

The finely regulated differentiation process is accompanied by progressive hemoglobinization and reorganization of membrane and cytoskeletal proteins. Erythroblast chromatin becomes progressively compacted starting from the basophilic stage. Orthochromatic erythroblasts exit the cell cycle, their transcriptional activity is significantly suppressed, except for the regions encoding important erythroid-specific genes, such as globin genes and erythroid cell membrane proteins [14]. Finally, orthochromatic erythroblasts expel their nuclei and form a pool of reticulocytes. A pyrenocyte containing a transcriptionally inactive nucleus surrounded by a thin rim of cytoplasm is engulfed by macrophages [15]. Reticulocytes enter the bloodstream and undergo further maturation steps, including membrane remodelling, autophagic organelle clearance and hemoglobin accumulation. Mature biconcave erythrocytes lack the Golgi apparatus, endoplasmic reticulum, mitochondria, and ribosomes [16,17].

Although all vertebrate erythroid cells undergo chromatin condensation, only mammalian erythrocytes are enucleated. Red cells in some non-mammalian vertebrates contain a special linker histone H5 [18,19,20]. Besides, unlike the nuclei of mammalian erythrocytes, their nuclei are not polarized. It is hypothesized that enucleation is a mammalian evolutionary adaptation to enhance blood cells circulation, making them more flexible and allowing them to pass through small capillaries. Nuclear expulsion is also assumed to provide more intracellular space for hemoglobin.

Ex vivo and in vitro culturing of erythroid progenitor cells have expanded our understanding of terminal erythropoiesis. Molecular and signaling pathways have been discovered that play an important role in chromatin condensation, enucleation and reticulocyte maturation. It is now clear that complex epigenetic regulation is involved in the control of transcriptional program in highly specialized erythroid cells, defining their differentiation trajectory.

## 3. Histone Post-Translational Modifications

Erythropoiesis is one of the most extensively characterized cell differentiation models, serving as a paradigm for understanding the mechanisms of development and differentiation in other cell types. Epigenetic studies of distinct erythroid populations have shown that chromatin states differ by specific differentiation stages, with clear transitions between some stages, which indicates that stage-specific activation of gene activity during erythropoiesis is a stepwise process involving complex cis-regulation [21,22,23,24,25].

Post-translational modifications of the flexible histone tails significantly contribute to the chromatin structure and stability. For instance, acetylation of histones by histone acetyltransferases (HATs) destabilizes closed chromatin structure, making DNA more accessible to transcription factors. In contrast, deacetylation of histones by histone deacetylases (HDACs) stabilizes chromatin by strengthening histone–DNA interaction, which is critical for heterochromatin formation. During erythroid maturation expression of histone acetyltransferase Gcn5 gradually decreases, which results in the drop of the global levels of the several histone acetylation markers, especially H3K9Ac, H4K5Ac, H4K12Ac, and H4K8Ac [26]. This is presumably regulated through protein levels of c-Myc that are downregulated beginning at the CFU-E stage and are reduced dramatically during the last stages of maturation. Ectopic expression of Gcn5 partially blocks chromatin condensation and subsequent enucleation, making chromatin conformation more open. Expression of c-Myc at physiological levels during the terminal stages of erythropoiesis also blocks chromatin condensation and enucleation [26].

Consistent with these data, HDAC2 level, which gradually increases during erythropoiesis, is critical for chromatin condensation and enucleation. Treatment with pharmacological inhibitors of HDAC2 or HDAC2 shRNA knockdown blocks chromatin condensation and enucleation of cultured mouse fetal erythroblasts [27]. In addition, HDAC6 plays an interesting role in the control of enucleation through deacetylation of the mDia2 formin protein [28]. mDia2 protein is essential for cell motility, cell polarity, vesicle trafficking, and cytokinesis, directly mediating actin assembly [29,30]. The mDia2-deficient erythroid cells have decreased accumulation of F-actin in the cleavage furrow during late differentiation from proerythroblasts and fail to complete cytokinesis [31]. These studies reveal the critical role of cooperative inhibition of histone acetylation and activation of histone deacetylation in condensation of chromatin in terminal erythroblasts [32]. Surprisingly, in contrast to these studies in mice, recent mass spectrometry data obtained on human erythroid cells did not reveal a global decrease in histone acetylation. There was a decline in H4 lysine 16 acetylation (H4K16Ac) level, associated with RNA polymerase II pause release, but at the same time, multiple acetylation marks including H3K36Ac and H3K23Ac increased during human erythropoiesis [23]. These results suggest that RNA Pol II pausing dynamics may be an important regulator of terminal erythroid maturation. 

Methylation of histone tails is also an important regulator of chromosome condensation. Setd8 (also known as KMT5A, PRSet7, and Set8) is the sole enzyme that can catalyze monomethylation of histone H4 lysine 20 (H4K20) [33]. Setd8 and H4K20me1 marker are regulated in a cell-cycle-specific manner. The expression of Setd8 reaches its peak in G2/M, where high H4K20me1 level promotes condensation of mitotic chromosomes. Their abundance reaches its minimum during S-phase due to ubiquitin-dependent degradation [34,35]. This methyltransferase is important for cell-cycle progression, chromatin condensation, and genomic stability and has recently been found to have an essential function during erythropoiesis [36]. Deletion of Setd8 in erythroid cells results in early embryonic lethality due to profound anemia. Setd8-deficient erythroblasts had significant phenotypic abnormalities, including impaired nuclear condensation and heterochromatin accumulation, defects in transcriptional repression and destruction of the nuclear membrane. The exact mechanisms by which Setd8 impacts chromatin condensation in erythroblasts remain elusive. It was shown that in both Setd8 knock-out (KO) flies and mice improper mitotic chromosome condensation is observed. Besides, embryonic fibroblasts from Setd8 KO mice demonstrate cell cycle defects, chromatin decondensation, and enlarged nuclei [37]. Finally, forced expression of Setd8 during S-phase in U2OS cells led to increased binding of CAP-D3 and CAP-G2, two subunits of chromatin compaction factor condensin II, to replicating chromosomes which resulted in aberrant pre-mitotic chromatin [35,38]. Presumably, this is because H4K20me1 is specifically recognized by condensin II component CAP-D3 [38]. These results suggest that expression dynamics of Setd8 throughout the erythroblast cell cycle may mediate chromatin condensation during terminal differentiation. 

Extrinsic signals can also induce changes in histone methylation status during erythroid cell maturation. Binding of EPO with EPO receptor activates Jak2-Stat5 signaling cascade. This appears to promote the expression of Suv420h2, a methyltransferase that adds methyl groups to mono-methylated K20 of histone H4 (methylated by Setd8) to form H4K20me3 [39]. Di- and tri-methyl marks on H4K20 have been reported to play roles in transcriptional pausing, chromatin condensation, and cell cycle control [40,41]. These processes are critical for terminal erythroid maturation.

Specific histone modifications are necessary for normal chromatin compaction and enucleation in erythroblasts. Collectively these findings suggest the importance of histone post-translational modifications in regulation of global chromatin dynamics during erythropoiesis.

## 4. Histone Release and Degradation, Nuclear Openings

One of the hallmark features of mammalian erythroid differentiation is the formation of nuclear openings—extensive areas in the nuclear envelope of early erythroblasts lacking the components of the lamina, nuclear pore complex, and nuclear membrane [42,43]. Distinct from mouse erythroblasts, where each cell contains a single opening, late-stage human erythroblasts tend to have multiple openings. The opening formation is mediated by caspase-3 cleavage of the nuclear lamina component lamin B protein. Importantly, histones are selectively released into the cytoplasm, which leads to the partial loss of the core and linker histones. However, the histone variant H2A.Z, as well as nuclear non-histone proteins involved in erythropoiesis, such as the GATA1 transcription factor and the HDAC2 enzyme, remain inside the nucleus, which indicates the selectivity of the process. Interestingly, H2A.Z histone variant is involved in the regulation of the dynamics of nucleosome arrangement [44], which suggests its possible role in chromatin reorganization during terminal erythropoiesis. Treatment of cultured erythroblasts with caspase inhibitor or caspase-3 knockdown blocks nuclear openings formation and inhibits histone release, chromatin condensation and enucleation. Cultured erythroblasts from caspase-3 knockout mice also exhibited a decreased number of nuclear opening events. Disruption of nuclear opening formation and histone release was found to play a critical role in the pathophysiology of red cell-related diseases such as myelodysplastic syndrome [43]. More recently, loss-of-function experiments in mouse primary erythroblasts, mouse erythroleukemia cells, and zebrafish revealed that Wdr26, the major subunit of the E3 ubiquitin ligase complex, mediates polyubiquitination and subsequent degradation of a fraction of nuclear proteins, including lamin B and H2A [45]. Wdr26-deficiency in erythroblasts impaired nuclear opening formation and chromatin condensation during differentiation. Importantly, caspase-mediated cleavage and Wdr26-dependent degradation are two independent pathways in differentiating erythroblasts. 

Histone release in erythroblasts is a unique example of selective transport carried out not through the nuclear pore complex, but through huge openings in the nucleus. The details of this mechanism, as well as why DNA is not released through these openings, are not yet clear. However, the process does not seem to be passive as not all major histones are released.

Another protein export pathway acting in late erythroid development is RAN transport machinery. Erythroid-specific nuclear export protein, Exportin 7 (Xpo7), may facilitate migration of some nuclear proteins, including core histones H2A and H3, to the cytoplasm [46]. Xpo7 knockdown severely disrupted normal chromatin condensation in differentiating erythroblasts. Taken together, these studies confirm the significance of histone release during chromatin condensation in terminal erythropoiesis. Yet, how the abovementioned large-scale changes in histone modifications and their composition affect the 3D regulatory landscape remains unrevealed.

## 5. Transcriptomic and Chromatin Accessibility Analysis

Differentiation from early erythroid precursor cells is characterized by a highly coordinated transcriptional program. It regulates dramatic alterations in their phenotypic features, such as metabolic shifts, decrease in cell size, hemoglobinization, membrane reorganization, and nuclear condensation [14,23,24,47]. This highlights that terminal erythroid differentiation is a unique process in which each cell division is simultaneously associated with differentiation. 

Stage-specific patterns of transcriptional activity in erythroid cells were described via global gene expression analysis in morphologically discrete FACS-sorted cell populations [48]. RNA was prepared from pure populations of human and murine erythroblasts at differing stages of terminal erythroid differentiation. One striking observation was a near-global decrease in gene expression in late murine erythroblasts, which, however, has not been observed on human cells. Importantly, in both human and mouse data, in the set of genes with high expression in proerythroblasts decreasing during differentiation, GO terms were enriched for differentially expressed genes related to protein synthesis and DNA metabolism including replication, repair, and cell cycle. It is worth mentioning that An et al. used an in vitro culturing system starting with CD34+ umbilical cord cells to obtain differentiated human erythroid cells [48]. However, some molecular traits in primary erythroblasts and in erythroblasts differentiated in vitro may differ. In more recent work gene expression dynamics from ProE to OrthoE was evaluated by scRNA-seq of primary terminally differentiated human erythroid cells, isolated directly from adult bone marrow and umbilical cord blood samples [24]. In addition to the well-known regulators of erythropoiesis, previously identified via bulk RNAseq in cultured erythroid cells, e.g., KLF1, GATA1, NFE2, BCL11A, LRF, TET2/3, and MYB [49,50], a set of novel regulatory genes have been recognized, that can potentially play a key role in regulating of terminal differentiation of primary human erythroid cells, especially erythroblast karyopyknosis and enucleation. In particular, AKAP8L gene, which was previously found to regulate nuclear envelope dynamics and histone deacetylation during G2/M transition [51,52]. Although Huang et al. scRNAseq results are largely consistent with previous works, they found that transcriptomic profiles of erythroid cells isolated directly from bone marrow show some differences [24]. Importantly, there was a global decrease in the number of genes expressed as erythroblast differentiation progressed, which is consistent with a decrease in the number of detected proteins [53]. At the same time, there was an increase in the expression of hemoglobin encoding genes and a set of genes that are associated with the regulation of hemoglobin synthesis, erythroblast structural remodeling and karyopyknosis.

Integration of transcriptomic patterns with dynamics of open chromatin landscape enabled researchers to more accurately understand the mechanisms that underlie HSC fate decisions. A detailed picture of the accessible chromatin landscape and transcription factor binding throughout human erythroid differentiation was obtained by Ludwig et al. [22]. They paired RNAseq and ATACseq to characterize the chromatin accessibility profile in the ex vivo differentiation system from human hematopoietic stem and progenitor cells. Cell cycle-associated processes showed maximum transcriptional activity at the ProE stage, which is concordant with the high proliferative capacity of these cells. The processes of heme biosynthesis and oxygen transport were the most active at the BasoE and PolyE stages, consistent with the peak of hemoglobin synthesis during these stages. Regulation of catabolic processes prevails during the OrthoE and Reticulocyte stages, as autophagic organelle clearance occurs before maturation into erythrocytes. Identified clusters of differential accessible peaks showed dynamic patterns similar to what was described for the transcriptomic profiles. Notably, even though PolyE and OrthoE stages are associated with global chromatin condensation and decreasing accessibility, a set of genomic regions was identified, showing maximum accessibility at these stages. The peaks specific for the late stages of differentiation were enriched in genes showing the highest levels of expression at the OrthoE stage, e.g., the TMCC2 locus, one of the most strongly induced genes during the terminal stages of erythropoiesis.

Coherence between the dynamic program of transcriptional activity during erythroid maturation and a tissue-specific regulatory DNA landscape are covered below.

## 6. DNA Demethylation

Changes in DNA methylation pattern play an important role in gene expression, parental imprinting, X chromosome inactivation, and transposable elements. Most promoters and enhancers generally remain unmethylated [54]. Data from mouse fetal liver erythroblasts and human umbilical cord-derived erythroid cells reveal that transition from erythroid progenitors to more mature erythroblasts is associated with significant demethylation [23,55,56]. Genome-wide loss of one-third of all DNA methylation at nearly all genomic loci within three cell divisions was reported by Shearstone et al. [57]. The most prominent decrease in methylation levels is observed between BFU-E and CFU-E stages, followed by a gain of methylation between CFU-E and proerythroblast stages. Further differentiation starting from the proerythroblast and up to the orthochromatic stage is associated with extensive genome hypomethylation. Although DNA hypomethylation encompasses the entire genome, it is most evident in gene bodies and intergenic regions [57]. This is consistent with the data on a positive correlation between the level of gene-body methylation with gene expression level [58]. However, it is worth noting a contradiction between the observed hypomethylation in promoters and the general silencing of transcription in erythroblasts.

The most prominent methylation losses were observed in regions with initially high methylation levels, such as non-CpG island promoters. Both de novo methyltransferases Dnmt3a and Dnmt3b were significantly down-regulated during differentiation along with upregulation of Gadd45a, a protein involved in active demethylation. Presumably, mechanisms leading to global demethylation may function to promote removal of methylation marks at sites of up-regulated erythroid genes, as Dnmt1 knockdown in fetal liver cells accelerates erythroid gene induction with differentiation.

Besides, a global loss in methylation was also detected in imprinted genes and transposable elements—regions that are stably methylated in somatic cells. However, at the same time, global demethylation was not accompanied by an increased transcription of transposable elements, as was shown for the LINE-1 retrotransposon [57].

The ten-eleven translocation (TET) family proteins have been recently documented to play a role in erythropoiesis [59]. The family consists of three enzymes, i.e., TET1, TET2, and TET3, that catalyze active DNA demethylation. TET proteins oxidize 5-methylcytosine (5mC) to 5-hydroxymethylcytosine (5hmC) and can further convert 5hmC to 5-formylcytosine (5fC) and then to 5-carboxycytosine (5caC). 5fC and 5caC can be repaired to cytosine through the base-excision repair pathway [60]. TET3 knockdown in cultured human erythroblasts resulted in increased apoptosis of late-stage erythroblasts and impaired enucleation. Bi/multinuclear late-stage erythroblasts were observed in a population of cells. In contrast, knockdown of TET2 resulted in delayed differentiation of progenitors with an increased number of CFU-E cells. Surprisingly, there were no significant changes in 5mC levels upon TET2/3 downregulation, except for the increased 5mC level on the orthochromatic stage. The overall 5hmC level decreases during the CFU-E-proerythroblast transition to an almost undetectable level and remains low during further differentiation [59]. 

## 7. Chromosome Conformation Studies

Tissue-specific chromatin folding patterns reflect different patterns of gene expression. How chromatin architecture and its modifications interrelate with gene activity as cell fate decisions are being made continues to be studied. Erythropoiesis represents a convenient model system for studying the molecular mechanisms of gene regulation within the same lineage during commitment. Alpha-globin locus represents a canonical example of developmental stage specific promotor-enhancer dynamics.

Chromosome conformation capture-based techniques, in particular Hi-C, have become a useful tool for analyzing the structural and functional relationships in the genome. This group of methods has played an important role in identifying the spatial context in which interactions of distal regulatory elements take place. Studies have revealed functional units of chromatin, including compartments, topologically associated domains (TAD), and chromatin loops [61,62,63]. Compartments A and B overlap with transcriptionally active and inactive chromatin regions, respectively, correlating with the distribution of histone marks, transcriptional activity, and different GC composition. At a more local level organization of chromosomes into TADs is thought to play a role in transcriptional regulation, facilitating enhancer-promoter interactions within a TAD and restricting such loops between neighboring TADs. It is argued that TADs function as insulation units, ensuring the selectivity of promoter-enhancer interactions [64,65]. TADs, isolated by CCCTC-binding factor (CTCF) binding, are considered stable among different cell types [66], evolutionarily conservative [66] and unchanged during differentiation [67], partially erased only during early embryogenesis in zygote [66] and completely disappearing in mitotic cells [68,69].

The relationship between chromatin structure and gene regulation was examined on the in vitro differentiated fetal and adult human erythroblasts [70]. The Hi-C results suggest that chromatin partitioning into A/B compartments, representing separation of active and repressed regions, is highly similar between fetal and adult erythroblasts. Only 5% of the genome switched compartments between the fetal and adult stages. At the submegabase level TAD structures were also strongly correlated between fetal and adult erythroblasts, suggesting global as well as local similarities in chromatin folding patterns between these stages. Closer inspection of the b-globin TAD was carried out using Capture-C and CTCF ChIPseq experiments. Human β-globin locus encompasses five coding genes specifically expressed in erythroid cells: the embryonic-specific ε-globin, fetal-restricted Gγ- and Aγ- globin, and adult δ- and β- globin genes. The sub-TAD structures surrounding the β-globin locus, demarcated by CTCF sites, are erythroid-specific, but invariant between developmental stages. However, the regulation of fetal-to-adult globin switching and silencing of fetal hemoglobin during development depends on the conformational switch of the HBBP1 region contacts. HBBP1 is a pseudogene located in the intergenic region between the Aγ- and δ-globin genes. And in adult cells the HBBP1 region interacts with the ε-globin gene, isolating the γ-globin from LCR (locus control region)—a strong upstream enhancer. Interactions of the LCR with the relevant β-type globin gene promoters are regulated in a developmental stage-specific manner [71]. This configuration of β-globin locus provides contacts of the LCR with adult-specific δ-globin and β-globin genes [70]. 

In Oudelaar et al. the Hi-C modification called Tiled-C in combination with ATAC and single-cell RNA-seq was used to study the relationship between the formation of structural chromatin interactions, the activation of regulatory elements and gene expression during development [72]. An important innovation of Tiled-C is optimization for small cell numbers. Therefore, the authors could isolate and analyze highly purified primary cells, thereby increasing the temporal resolution through in vivo mouse erythroid differentiation. The authors analyzed the chromatin architecture of several key erythroid genes through the in vivo stages of erythroid differentiation in mice, starting from the HSPCs. The analysis was specifically focused on the α-globin genes, as the regulatory elements of this locus are extremely well characterized. They found that TAD, comprising the α-globin genes, is already established in hematopoietic stem cells and is maintained during differentiation. Which is in agreement with previous works, showing that TADs are relatively stable units during differentiation and development [73,74,75]. Subsequent chromatin reorganization in differentiating cells involves the formation of smaller domains (sub-TADs) within TADs, strengthening specific enhancer-promoter interactions. These structural changes are concomitant with an upregulation of gene activity. In contrast to the literature [76], they found that formation of these enhancer-promoter interactions does not precede gene activation, but occurs simultaneously with the gradual activation of alpha-globin expression. Importantly, the same is true for other erythroid gene loci that were examined [72].

Experiments on chicken HD3 cells have revealed that erythroid-specific transcription outburst of the alpha-globin genes has a strong impact on adjacent non-related genes and chromatin structure of the encompassing TAD. Ulianov et al. have analyzed a 2.7 Mb region of chicken chromosome 14 comprising an alpha-globin gene domain, a set of non-globin genes, and a gene desert [77]. It turned out that along with alpha-globin transcription burst in terminally differentiated erythroblasts, an increased transcription of several adjacent housekeeping genes and intergenic regions is observed. This is accompanied by local changes in chromatin structure within the chromosome region encompassing the alpha-globin gene cluster. While a 300 Kb region around the alpha-globin gene cluster becomes decompacted in differentiated erythroid cells, the alpha-globin cluster sub-TAD becomes highly enriched in chromatin interactions.

Exploration of the tissue-specific organization of α-globin locus has been expanded in work by Chiariello et al. Using Capture-C data from Oudelaar et al. for ESCs and erythroid cells, they found that a striking pattern of chromatin contacts is formed in the latter [78,79]. While in ESCs the locus is organized into a large domain, with no preferential contacts between its regulatory elements and a high degree of intermingling with its flanking regions, in erythroid cells α-globin genes are organized into a separate self-interacting domain with highly specific contacts. On the erythroid Capture-C matrix this interaction-enriched region is stretched along the antidiagonal direction of the Hi-C matrix and comprises regions flanking the α-globin genes. Notably, the emerging structure does not fit the definition of a TAD or a loop domain, associated, respectively, with a “square” or “dot” pattern on the contact map. Applying the polymer modeling approach, the authors suggested that the antidiagonal pattern could be explained by a hairpin-like organization. Importantly, the folding of the α-globin locus most likely is driven by the thermodynamics of microphase separation, whereas CTCF / cohesin interactions only marginally participate in this process. 

The Hi-C method was used to study global chromatin dynamics in chicken erythrocytes [80]. One of the remarkable features of chromatin in erythrocytes was the absence of TADs, which are considered stable units of the interphase genome organization. One of the possible explanations for the disappearance of these structures is the unusual distribution of the architectural protein CTCF in the nuclei of chicken erythrocytes. In mature erythrocytes, CTCF seems to be excluded from chromatin and is accumulated in discrete nuclear bodies [81]. In addition to the absence of loop domains, a global rearrangement of genomic contacts was also observed on a larger scale. An increased contact frequency for genomic loci separated by 15 Mb is displayed as a second diagonal band on erythrocytes’ Hi-C maps. As a general rule in the interphase nucleus the probability of contacts between loci tends to decrease as distance increases. A P(s) plot, resembling a probability of chromatin contacts depending on genomic distance between loci, confirmed the presence of the local peak of contact frequencies at ~15 Mb in erythrocytes. This pattern of chromatin interactions resembles the pattern of contacts obtained by Gibcus et al. in metaphase chromosomes [68]. The authors also indicated a global loss of topological domains and smaller loops [82]. Moreover, erythrocyte chromosome organization lacking TADs and loops along the diagonal, with an interaction pattern manifesting as a second diagonal on a Hi-C map is not unique for avian erythrocytes, but is evolutionary conserved [83]. A set of Hi-C data on mature erythrocytes from birds, reptiles, amphibians, bony fish and mammalian erythroblasts was analyzed. Chromatin in erythrocytes of non-mammalian species and late erythroblasts of mammals is organized in a fundamentally different way than the usual interphase nucleus. It lacks a hierarchical genomic organization at the TAD level. Besides, contact maps of erythroid cells from mammals, birds, reptiles and amphibians demonstrate an increased contact frequency at a distance of 10–30 Mb. 

Another Hi-C related study expands the knowledge on chromatin dynamics during differentiation. In work by Zhang et al. a low-input Hi-C variant, tagHi-C, was used to obtain contact maps for 10 major cell types of the hematopoietic lineage [84]. Contact maps from some cell types of the myeloid line, namely granulocytes (GR), megakaryocytes (MK) and megakaryocyte-erythrocyte progenitors (MEP), displayed an unusual distribution of chromatin contacts as evidenced by the shape of P(s) curve. Compared to HSCs and earlier progenitors, the frequency of chromatin contacts in these cells was higher at 10 Mb, with a more rapid decline at 30 Mb. Decreased frequency of long-range chromatin interactions in MEP, MK, and GR cells resembles the Hi-C maps of condensed chromosome structures reported for mitotic cells [85]. To further explore the chromatin structure 3D chromosomal models were created for each cell type. Surprisingly, a more elongated chromosomal shape in MEP, MK and GR was observed, indicating a more condensed chromatin conformation in these cell types. Subsequent MNase susceptibility assay confirmed the condensed chromatin conformation. In addition, the tagHi-C data and FISH have shown that at the stage of common lymphoid/common myeloid progenitor chromosomes acquire the Rabl configuration, and this configuration gradually becomes more common during differentiation of blood cells. In particular, MEP and GR cells showed a strong tendency to adopt the Rabl configuration with strong centromere and telomere clustering.

## 8. Conclusions

Mammalian erythrocytes are highly specialized cells: after lineage commitment, progenitors undergo four to five cell cycles before enucleation. Their morphology changes dramatically during this process, occurring in a relatively short time frame. One of the hallmarks of erythroid differentiation, along with hemoglobinization and autophagic organelle clearance, is nuclear condensation and extrusion. Chromatin condensation associated with transcriptional silencing is a necessary requirement for erythroid enucleation. In this review, we have covered some ways in which chromatin dynamics during erythropoiesis is regulated. In particular, open areas are formed in the nuclear envelope of early erythroblasts, lacking components of the lamina and nuclear membrane. The dynamic nuclear opening is required for the selective release of a fraction of core and linker histones [42]. Remaining histones undergo deacetylation, which facilitates chromatin condensation. Analysis of the relationships between gene expression, epigenome, and the global three-dimensional organization of the genome may become a way to further study the mechanisms of nuclear condensation during erythroid differentiation. With the development of single-cell approaches, including studies of the three-dimensional genome, one should expect a more mechanistic understanding of how the interactions of individual regions, as well as the structure of chromatin as a whole, change at each stage of erythroid differentiation.

## Data Availability

Not applicable.

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
