# Peer review of "Genome Reorganization during Erythroid Differentiation"

_genes, 2021, doi:10.3390/genes12071012_

Round 1

Reviewer 1 Report

The authors give a comprehensive and detailed overview of the erythropoiesis on the molecular level, focusing on chromatin reorganization, detailing steps from erythroid progenitors to enucleated reticulocytes. Authors discuss histone modifications, such as acetylation and methylation with opposing effects on chromatin, histone release, nuclear degradation, transcription and open chromatin levels, DNA methylation, and finally 3D chromosomal organization. The review is well written, but English language should be improved.

 Comments:

"such the chromatin organization",

remove the

"It is now clear that such the chromatin organization of erythrocytes is an extremely rare case, involving histone modifications, e.g. deacetylation, a low content of non-histone proteins in the nucleus, and probably other yet unrevealed factors",

the sentence is difficult to understand, what is extremely rare among these, rephrase

"Although in all vertebrates erythroid cells undergo chromatin condensation"

remove "in"

"acetylation of histones by histone acetyltransferases (HATs) destabilizes chromatin"

destabilizes closed chromatin structure, not chromatin itself

"It cannot be ruled out that some molecular 201 and cellular traits in primary erythroblasts in vivo and in erythroblasts generated in vitro 202 may differ."

Sentence redundant 

"... a set of novel regulatory genes have been recognized, that can potentially play a key role in regulating of terminal differentiation of primary human erythroid cells"

authors do not mention which particular genes these are that have been found or at least few examples

"Although Huang et al. ..."

"It is worth mentioning that An et al...."

the authors do not provide reference numbers, unclear what citation this refers to

"Notably, a set of regulatory elements were identified showing maximum accessibility specific to PolyE and OrthoE stages, which are associated with chromatin condensation and decreasing accessibility"

How many elements, authors should expand on this. were they tested for their function in decreasing chromatin accessibility

Authors should expand on the DNA demethylation chapter, is DNA methylation increased in later stages?

"they found that formation of these enhancer-promoter interactions does not precede gene activation, but occurs simultaneously with the gradual activation of alpha-globin expression"

Authors should expand on how did they infer synchronicity.

"One of the possible explanations for the disappearance of these structures is the unusual distribution of the architectural protein CTCF in the nuclei of chicken erythrocytes"

Authors should expand on the distribution of ctcf

Author Response

Dear Editor,

We are grateful to the editors and reviewers for their time and constructive comments on our manuscript. We have implemented their comments and suggestions and wish to submit a revised version of the manuscript for further consideration in the Genes journal. Below, we also provide a point-by-point response explaining how we have addressed each of the reviewers' comments.

We look forward to the outcome of your assessment.

Yours sincerely,

Nariman Battulin

Reviewer 1

The authors give a comprehensive and detailed overview of the erythropoiesis on the molecular level, focusing on chromatin reorganization, detailing steps from erythroid progenitors to enucleated reticulocytes. Authors discuss histone modifications, such as acetylation and methylation with opposing effects on chromatin, histone release, nuclear degradation, transcription and open chromatin levels, DNA methylation, and finally 3D chromosomal organization. The review is well written, but English language should be improved.

 Comments:

"such the chromatin organization",

remove the

Thank you for this suggestion. Done.

"It is now clear that such the chromatin organization of erythrocytes is an extremely rare case, involving histone modifications, e.g. deacetylation, a low content of non-histone proteins in the nucleus, and probably other yet unrevealed factors",

the sentence is difficult to understand, what is extremely rare among these, rephrase

Thank you for this suggestion. We have rephrased the sentence, now it is:

«It is now clear that chromatin organization of erythrocytes is an extremely rare case. Their nuclei become gradually more condensed during maturation, which results in extensive transcriptional silencing. Chromatin reorganization involves histone modifications, e.g. deacetylation, DNA demethylation, selective histone release and probably other yet unrevealed factors.»

"Although in all vertebrates erythroid cells undergo chromatin condensation"

remove "in"

Thank you for this suggestion. Done.

"acetylation of histones by histone acetyltransferases (HATs) destabilizes chromatin"

destabilizes closed chromatin structure, not chromatin itself

Thank you for this suggestion. Done.

"It cannot be ruled out that some molecular 201 and cellular traits in primary erythroblasts in vivo and in erythroblasts generated in vitro 202 may differ."

Sentence redundant 

Thank you for this suggestion. We have rephrased the sentence, now it is:

“However, some molecular traits in primary erythroblasts and in erythroblasts differentiated in vitro may differ.”

"... a set of novel regulatory genes have been recognized, that can potentially play a key role in regulating of terminal differentiation of primary human erythroid cells"

authors do not mention which particular genes these are that have been found or at least few examples

Thank you for this suggestion. We have added following sentence

“In particular, AKAP8L gene, which was previously found to regulate nuclear envelope dynamics and histone deacetylation during G2/M transition [1,2]

"Although Huang et al. ..."

"It is worth mentioning that An et al...."

the authors do not provide reference numbers, unclear what citation this refers to

Thank you for this suggestion. Fixed.

"Notably, a set of regulatory elements were identified showing maximum accessibility specific to PolyE and OrthoE stages, which are associated with chromatin condensation and decreasing accessibility"

How many elements, authors should expand on this. were they tested for their function in decreasing chromatin accessibility

I'm afraid that the formulation was not accurate enough. These elements do not result in chromatin condensation. We have paraphrased this sentence as follows:

“Notably, even though PolyE and OrthoE stages are associated with global chromatin condensation and decreasing accessibility, a set of genomic regions was identified showing maximum accessibility at these stages.”

Authors should expand on the DNA demethylation chapter, is DNA methylation increased in later stages?

Thank you for this suggestion. We have expanded the discussion of this question by adding the following piece:

“The most prominent decrease in methylation levels is observed between BFU-E and CFU-E stages, followed by a gain of methylation between CFU-E and proerythroblast stages.  Further differentiation starting from the proerythroblast and up to the orthochromatic stage is associated with extensive hypomethylation of the genome. Although DNA hypomethylation encompasses the entire genome, it is most evident in gene bodies and intergenic regions [55]. This is consistent with the data on a positive correlation between the level of gene-body methylation with gene expression level [3]. However, it is worth noting a contradiction between the observed hypomethylation in promoters and the general silencing of transcription in erythroblasts.”

"they found that formation of these enhancer-promoter interactions does not precede gene activation, but occurs simultaneously with the gradual activation of alpha-globin expression"

Authors should expand on how did they infer synchronicity.

Thank you for this suggestion. We added following sentence for this:

“An important innovation of Tiled-C is optimization for small cell numbers. Therefore, the authors could isolate and analyze highly purified primary cells, thereby increasing the temporal resolution through in vivo mouse erythroid differentiation.”

"One of the possible explanations for the disappearance of these structures is the unusual distribution of the architectural protein CTCF in the nuclei of chicken erythrocytes"

Authors should expand on the distribution of ctcf

Thank you for this suggestion. We expand on distribution of CTCF in chicken erythrocytes.

“One of the possible explanations for the disappearance of these structures is the unusual distribution of the architectural protein CTCF in the nuclei of chicken erythrocytes. In mature erythrocytes, CTCF seems to be excluded from chromatin and is accumulated in discrete nuclear bodies [76].”

Reviewer 2

In this review article the authors provide a comprehensive overview on chromatin reorganization during erythroid differentiation.Thereby, the authors look at changes of chromatin and its conformation from different perspectives and at different levels of resolution. The article is well structured and written.

Some minor comments/questions: 

The authors describe changes of histone aceytlation and loss of lamins during erythroid differentiation. Actually, previous publications dealing with other cell types have described a correlation of histone hypoactylation and lamina associated domains. Does this tight link of histone acetylation state and lamin association impact histone release and nuclear opening?

The authors mention that regulation of DNA methylation plays a role in erythroid differentiation, in particular downregulation of DNA methyltransferases promoting passive DNA demethylation and upregulation of GADD45 triggering active DNA demethylation. What´s about the members of the TET family, which are the enzymes responsible for converting methylcytosine to hydroxymethylcytosine? Changed expression levels of TET2 and TET3 are mentioned in the chapter : "Transcriptomic and chromatin accessibility analysis", but not when it comes to describing the process of active DNA demethylation. 

Thank you for this suggestion. We have expanded the discussion of this question by adding the following piece:

“The ten-eleven translocation (TET) family proteins have been recently documented to play a role in erythropoiesis [4]. The family consists of three enzymes, i.e., TET1, TET2, and TET3, that catalyze active DNA demethylation. TET proteins oxidize 5-methylcytosine (5mC) to 5-hydroxymethylcytosine (5hmC) and can further convert 5hmC to 5-formylcytosine (5fC) and then to 5-carboxycytosine (5caC). 5fC and 5caC can be repaired to cytosine through the base-excision repair pathway [5]. TET3 knockdown in cultured human erythroblasts resulted in increased apoptosis of late-stage erythroblasts and impaired enucleation. Bi/multinuclear late-stage erythroblasts were observed in a population of cells. In contrast, knockdown of TET2 resulted in delayed differentiation of progenitors with an increased number of CFU-E cells. Surprisingly, there were no significant changes in 5mC levels upon TET2/3 downregulation, except for the increased 5mC level on the orthochromatic stage. Overall 5hmC level decreases during the CFU-E-proerythroblast transition to an almost undetectable level and remains low during further differentiation [4].”

In this context: is it correct to say DNA methylation plays a role in transposable elements? Actually, the role of DNA methylation is to suppress the expression of transposable elements, e.g. LINEs?

Thank you for this suggestion. We have expanded the discussion of this question by adding the following piece:

“Besides, a global loss in methylation was also detected in imprinted genes and transposable elements – regions that are stably methylated in somatic cells. However, at the same time, global demethylation was not accompanied by an increased transcription of transposable elements, as was shown for the LINE-1 retrotransposon [55].”

The authors describe the loss of TADs and mention the structural resemblances to metaphase chromosomes. What´s about the inactive X chromosome, which lacks usual TADs, but instead is organized in two super domains.

This is a great question. The main difficulty is that to answer it, it is necessary to distinguish between two X-chromosomes. For example, by use mice hybrids for obtaining erythroblasts. Unfortunately, our laboratory does not have such data, and we do not know the works with which such a design.

There is a review article in the journal VAVILOVSKIJ ŽURNAL GENETIKI I SELEKCII (2019-02-01) by the same two authors (and one additional one) with the title: Reorganization of chromatin during erythroid differentiation. Since the article is not open source and I am not familiar with the russian language I can not judge the extent of overlap with the content of the current article. Maybe the authors could comment on that

The review in the VAVILOVSKIJ ŽURNAL journal was aimed at non-specialists in the field of 3D genomics (primarily for students), so about half of it was devoted to the general principles of the three-dimensional organization of the genome.

About one page of text is devoted directly to erythroid cells.

The new review is a much more complete analysis of epigenetic events and their relationship to the three-dimensional architecture of the erythroid genome. This can be seen from the amount of cited literature. There are 85 references in the new review, 26 in the review from the VAVILOVSKIJ ŽURNAL.

The above review is open access at

https://vavilov.elpub.ru/jour/article/view/1874/1185

I confirm that the new review is not an English translation of the earlier Russian review article despite similar titles.

Reviewer 2 Report

In this review article the authors provide a comprehensive overview on chromatin reorganization during erythroid differentiation.Thereby, the authors look at changes of chromatin and its conformation from different perspectives and at different levels of resolution. The article is well structured and written.

Some minor comments/questions: 

The authors describe changes of histone aceytlation and loss of lamins during erythroid differentiation. Actually, previous publications dealing with other cell types have described a correlation of histone hypoactylation and lamina associated domains. Does this tight link of histone acetylation state and lamin association impact histone release and nuclear opening?

The authors mention that regulation of DNA methylation plays a role in erythroid differentiation, in particular downregulation of DNA methyltransferases promoting passive DNA demethylation and upregulation of GADD45 triggering active DNA demethylation. What´s about the members of the TET family, which are the enzymes responsible for converting methylcytosine to hydroxymethylcytosine? Changed expression levels of TET2 and TET3 are mentioned in the chapter : "Transcriptomic and chromatin accessibility analysis", but not when it comes to describing the process of active DNA demethylation. 

In this context: is it correct to say DNA methylation plays a role in transposable elements? Actually, the role of DNA methylation is to suppress the expression of transposable elements, e.g. LINEs?

The authors describe the loss of TADs and mention the structural resemblances to metaphase chromosomes. What´s about the inactive X chromosome, which lacks usual TADs, but instead is organized in two super domains.

There is a review article in the journal VAVILOVSKIJ ŽURNAL GENETIKI I SELEKCII (2019-02-01) by the same two authors (and one additional one) with the title: Reorganization of chromatin during erythroid differentiation. Since the article is not open source and I am not familiar with the russian language I can not judge the extent of overlap with the content of the current article. Maybe the authors could comment on that.

Author Response

(The authors gave the same response as above.)
